# High-Accuracy Clock Offsets Estimation Strategy of BDS-3 Using Multi-Source Observations

**Jianhua Yang [1,2], Chengpan Tang [1], Sanshi Zhou [1], Yezhi Song [1], Jinhuo Liu [1,2], Yu Xiang [3], Yuchen Liu [1,2], Qiuning Tian [4], Yufei Yang [3], Zuo Yang [3], Xiaogong Hu [1,*]**

[1] Shanghai Astronomical Observatory, Chinese Academy of Sciences, Shanghai 200030, China
[2] University of Chinese Academy of Sciences, Beijing 100049, China
[3] Beijing Satellite Navigation Center, Beijing 100094, China
[4] Research Institute of Telemetry, Beijing 100076, China
[*] Correspondence: hxg@shao.ac.cn

**Abstract:** Satellite clock offsets are the critical parameters for The Global Navigation Satellite Systems (GNSSs) to provide position and timing (PNT) service. Unlike other GNSSs, BDS-3 uses the two-way superimposition strategy to measure satellite clock offsets. However, affected by some deficiencies of the two-way superimposition strategy, the accuracy of BDS-3 clock offsets parameters is 1.29 ns (RMS), which is the main bottleneck for BDS-3 to improve its space signal accuracy. After analyzing problems in the clock offsets measurement process of BDS-3, the paper proposes a new strategy to real-time estimate high-accuracy satellite clock offsets. The clock offsets estimated by the new strategy show a good consistency with GBM clock offsets. The averaged STD of their differences in MEO is 0.14 ns, and the clock offsets estimated by the new strategy present less fluctuation in the 1-day fitting residuals. Applying the new clock offsets to prediction, BDS-3 can reduce its clock offsets errors from 1.05 ns to 0.29 ns (RMS), about 72%. The above results indicate that the new clock offsets estimated strategy can improve the accuracy of clock offsets parameters of BDS-3 effectively.

**Keywords:** BDS-3; clock offsets estimated; inter-satellite links; two-way satellite time and frequency transfer

## 1. Introduction

Because GNSSs measure distances and clock offsets by comparing clocks, GNSSs need to provide accurate messages, including their space and time. For BDS-3 open service, BDS-3 calculates the position and velocity of satellites in the Beidou Coordinate System (BDCS) [1], and the satellite clock offsets with respect to Beidou Time (BDT) [2,3] at first. After that, BDS-3 parameterizes the satellite's orbit and clock offsets to ephemeris and clock offset parameters and uploads them to the satellite for broadcasting [4,5]. The update frequency of the above messages are often every hour and the predicted time for users to use them is less than 2 hours. The influence of their errors on ground users is presented as signal-in-space errors (SISRE) [6,7]. SISRE is one of the most significant error sources in PNT, which consist of broadcast ephemeris errors and clock offsets parameters errors. Galileo performs the lowest SISRE in "GNSS Big 4" at present [8]. The BDS-3 constellation consists of 24 MEOs, 3 IGSOs, and 3 GEOs, whose SISRE is second to Galileo [9,10]. Statistical results show that the broadcast ephemeris errors can reach centimeter levels [11,12], which is at the top level in "GNSS Big 4" [8–10]. However, the clock offsets parameters errors of BDS-3 are 0.35 m (RMS) with respect to BDT, and evaluated by IGS precise clock offsets are more than 0.51 ns, which is larger than the errors of 0.14 m of Galileo. Therefore, the main reason why SISRE of BDS-3 is larger than Galileo is that the clock offsets parameters of BDS-3 with larger errors [9,10].

Generally, GNSS satellite clock offsets are the differences between the satellite atomic clock onboard and GNSS system time in GNSS conventional reference system. Their measurement is the most crucial step in GNSS generating satellite clock offsets parameters. Orbit determination and time synchronization (ODTS) is GNSS's most proven and widely used clock offsets estimation technology [13,14]. The ODTS can estimate satellite clock offsets simultaneously when estimating satellite orbit using GNSS downlinks pseudorange and carrier phase measurements from dozens to hundreds of global GNSS stations. For BDS-3, some results suggest that the clock offsets differences between real-time and batch-processed solutions in ODTS are less than 0.15 ns, which is as the same level as other GNSS [15]. The ODTS estimation technology has high requirements for GNSS station distribution and quantity. However, since the BDS monitor station are only distributed within China, it is difficult for BDS to realize above requirements. Under the above circumstances, BDS-3 uses the two-way comparison technology to measure satellite clock offsets [16]. The fundamental principle of the two-way comparison is to subtract two-way measurements to eliminate most of the errors related to spatial information, and obtain their clock offsets. As shown in Figure 1, the BDS-3 time synchronization system owns the Master Control Station (MCS) and Time Synchronization Stations (TSS). All of them are equipped with C-band antennas, allowing BDS-3 to measure their clock offsets by two-way measurements [17]. Moreover, the satellite and BDS-3 ground station are also equipped with L-band two-way antennas and Ka-band phased-array antennas. The above antennas allow BDS-3 to calculate L-band two-way clock offsets and Ka-band two-way inter-satellite clock offsets [18]. According to the tracking status of the ground antenna to satellite and the link plan of inter-satellite link (ISL) [19], BDS-3 superimposes the above clock offsets to obtain satellite clock offsets with respect to BDT. This strategy we named the two-way superimposition strategy. Although the two-way superimposition strategy can measure satellite clock offsets, whether overseas or intraregional, it puts forward high requirements for the ability and quantity of ground antenna. More importantly, the final satellite clock offset is only obtained by several superimpositions of the above two-way clock offset. Each superimposition of two-way clock offset is accompanied with accuracy loss and will decrease the accuracy of satellite clock offset parameters. As a result, some research finds that the clock offset parameters of BDS-3 are not accurate enough [9, 10]. With the continuous improvements and developments in terms of performance, availability, modernization, application, and hybridizing for current GNSS [20–22], BDS must face up to the problem in its clock measurements and take it as the guide to study the new approach to improve its signal-in space accuracy.

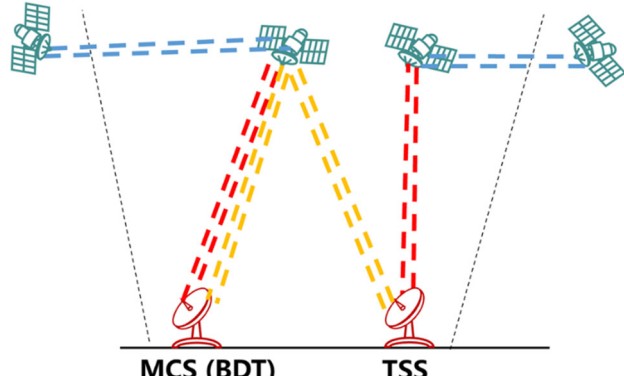

**Figure 1.** The construction of BDS time synchronization and the basic two-way measurements of BDS-3. Wherein, the red, blue, yellow curves are L-band satellite-station two-way measurements, Ka-band inter-satellite measurements, C-band inter-station two-way measurements, respectively.

For the above reason, we analyze the problems in the BDS-3 two-way superimposition strategy in the first part. Then, guided by solving the problems, we proposed a new

strategy. The new strategy considers and combines BDS-3 constellation composition and multi-source measurements accuracy to real-time estimate BDS-3 satellite clock offsets with higher accuracy. We analyze the accuracy and improvements of the new strategy in the last part.

## 2. Materials and Methods

### 2.1. Two-Way Superimposition Strategy

2.1.1. Fundamental Principles Process

BDS-3 owns a time synchronization, which is independent of satellite orbit determination. The fundamental measurements of BDS-3 two measurements include L-band two-way satellite-station measurements, C-band two-way inter-station measurements, and Ka-band inter-satellite measurements.

The fundamental equation for using two-way measurements to calculate two-way clock offsets is given as Equation (1) [15]:

$$clk_{BA}(t_0) = \left[ \left( \rho_{AB}(t_0) - r_{AB}(t_0) \right) - \left( \rho_{BA}(t_0) - r_{BA}(t_0) \right) \right] / 2 + \sum corr + Err + \varepsilon \tag{1}$$

where, $clk(t_0)_{BA}$ is the clock offsets of $A$ with respect to $B$; $\rho_{BA}(t_0)$ and $\rho_{AB}(t_0)$ are the measurement values $r_{AB}(t_0)$ and $r_{BA}(t_0)$ are the signal propagation distance between two atomic clock $A$ and $B$. $\sum corr$ is all kinds of errors correction in two-way measurements. They include phase center correction, relativity correction [23], time delay correction [24]. For L-band and C-band measurements, their signal passes through the ionosphere and troposphere, So, the errors correction of L-band and C-band measurements also include troposphere delay correction and ionosphere delay correction [25]. *Err* are the unknown residual errors. $\varepsilon$ is two-way noise.

Considering all satellite clock offsets with respect to BDT. BDS-3 needs to use the two-way superimposition strategy to calculate final satellite clock offsets with respect to BDT. The procedure of the two-way superimposition strategy is given in Figure 2. For the intraregional satellites are tracked by MCS antennas, BDS-3 can directly obtain the clock offsets by the L-band two-way satellite-station clock offsets of MCS. We present it as L in Figures 2 and 3. For the satellites are tracked by TSS antennas, the two-way superimposition strategy subs the TSS's C-band inter-station clock offsets and L-band two-way satellite-station clock offsets to obtain the final satellite clock offsets. We present it as L + C in Figures 2 and 3. For the overseas satellites, the two-way superimposition strategy subs the two-way clock offsets of intraregional satellite and Ka-band inter-satellite clock offsets to obtain the overseas satellite two-way clock offsets. We present it as L + C + Ka or L + Ka in Figures 2 and 3. According to the antenna track messages and ISL plans [23,24], the two-way superimposition strategy can real-time calculate all satellite's two-way clock offsets [19]. Then, by 3-day comparison, BDS-3 estimate and deduct the bias between the two-way clock offsets and GNSS L-band downlinks offsets. Last, the final clock offset parameters of BDS-3 can be fitted by the linear model.

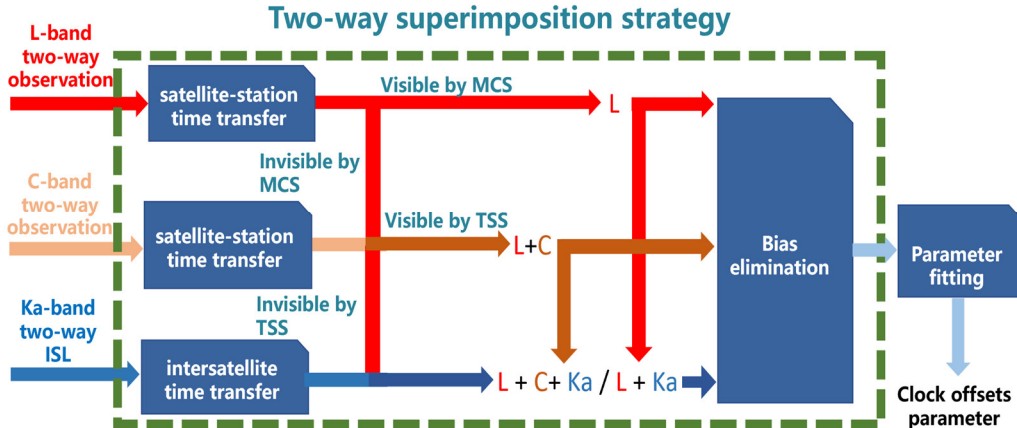

**Figure 2.** The procedure of the BDS-3 two-way superimposition strategy.

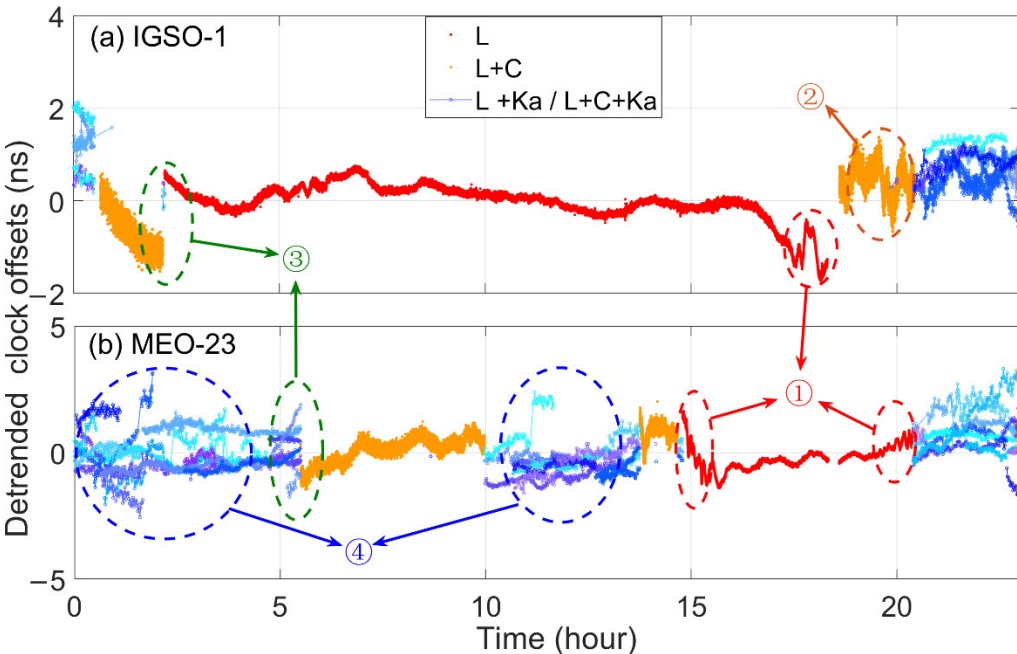

**Figure 3.** The typical one day detrended two-way clock offsets of IGSO and MEO measured by BDS-3 two-way superimposition strategy. ①–④ describe the problems and abnormal phenomena in the two-way superimposition strategy.

2.1.2. Problems

Taking the IGSO-1 and MEO-1 as examples, the detrended 24 h two-way clock offsets calculated by the two-way superimposition strategy are plotted in Figure 3.

In Figure 3. the red curves are the clock offsets only calculated by L-band two-way clock. The orange curves are the clock offsets calculated by C-band inter-station two-way clock offsets and L-band two-way clock offsets. The blue curves are the clock offsets through Ka-band two-way inter-satellite clock offsets, and the blue curves with different depths represent the clock offsets calculated through different intraregional satellites. Figure 3 demonstrates that whatever satellite is or is not over China. The two-way superimposition strategy can always measure their clock offsets. However, this figure also presents its shortcoming and problems:

- Because there is little difference between the L-band uplinks and the L-band downlinks, Equation (1) cannot effectively eliminate the ionosphere delay and multipath effects. When the elevation of the L-band antenna is not low enough, the above

residual errors will increase rapidly and decrease the accuracy of L-band two-way clock offsets (① in Figure 3).

- Because the noise of C-band measurements is far from other bands' measurements. The noise of clock offsets will become large when the satellite clock offsets are calculated through C-band two-way inter-stations clock offsets (② in Figure 3).
- Affected by temperature, antenna attitude, illumination, etc., the time delay of antennas has fluctuations, which are difficult to predict. Therefore, once the tracking antenna changes, the "step errors" will appear in the clock offsets (③ in Figure 3).
- The BDS-3 follows a time-division multiple access (TDMA) structure. One satellite connects with different visible satellites at different time in one connectivity cycle. Because an overseas satellite establishes links with different satellites frequently, the different "step errors" will be reflected in the clock offsets of overseas satellites, which are like the noise (④ in Figure 3).

The above reasons lead to poor accuracy of BDS-3 clock offsets, which is the main bottleneck limiting BDS-3 from improving itself signal-in-space accuracy. Therefore, based on the BDS-3 multi-source measurements, BDS-3 needs to urgently design a new strategy to measure the satellite clock offsets.

*2.2. The New Strategy*

From the above analysis, it can be seen that the reason why the accuracy of two-way clock offsets is poor is that the traditional strategy totally believes in the single two-way measurement and ignores that the errors of each measurement will also be superimposed in the final two-way clock offsets. All BDS-3 satellites are equipped with high-performance onboard atomic clocks [26–28], and the output signal has high stability in the short timescale. Therefore, the new strategy should believe in the ability of onboard atomic clocks instead of signal measurement. More importantly, unlike the traditional strategy, the new strategy abandons the C-band inter-station two-way clock offsets and L-band two-way clock offsets of MEOs and IGSOs, instead only using Ka-band inter-satellite between all satellites and L-band two-way clock offsets of GEOs. After the above changes, the new clock offsets estimated strategy includes three steps, shown in Figure 4.

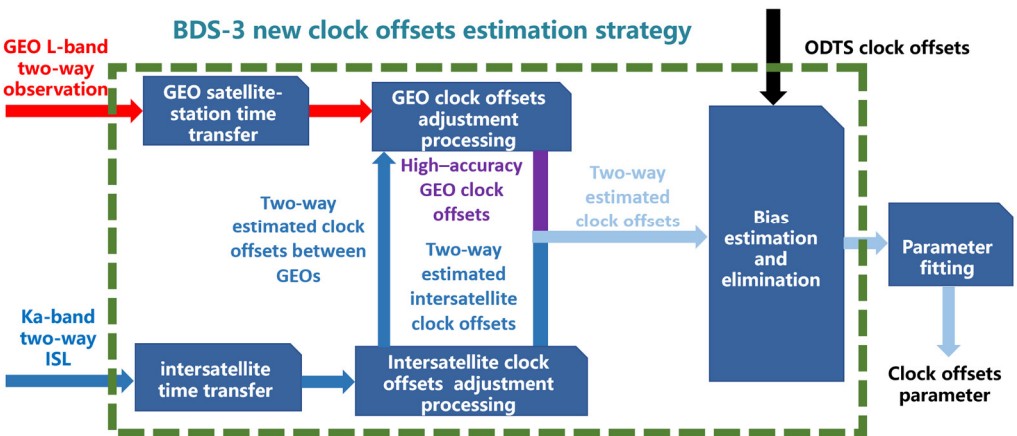

**Figure 4.** The procedure of the BDS-3 new clock offsets estimation strategy.

Firstly, the new strategy takes redundant Ka-band two-way inter-satellite clock offsets as the inputs and uses least square estimation to filter the noise and errors of the two-way clock offsets to estimate inter-satellite clock offsets with higher accuracy between all satellites, which is recorded as two-way estimated inter-satellite clock offsets. Then, taking the two-way estimated inter-satellite clock offsets between GEOs and their L-band two-way clock offsets as inputs, the new strategy further uses least square estimation to estimate one GEO's clock offsets with respect to BDT. Last, by making the differences between the ODTS clock offsets and two-way estimated clock offsets and averaging their

differences, the bias between two kinds of clock offsets can be calibrated and thereby realize the alignment of the two-way estimated clock offsets to the ODTS clock offsets. The following subsection will introduce each step of the new clock offsets estimation strategy of BDS-3.

### 2.2.1. Estimation of Inter-Satellite Clock Offsets

As shown in Figure 5a, BDS-3 follows the TDMA structure. One satellite connects with different satellites at different intervals to perform dual, one-way ranging measurements. In one connectivity cycle whose duration is 60 seconds, multiple Ka-band two-way inter-satellite clock offsets between one specific satellite and those on other satellites can be obtained by processing measurements using Equation (1). In fact, BDS-3 onboard atomic clocks perform high stability in the short timescale. The above redundant Ka-band two-way inter-satellite clock offsets can be further reduced to the same time (see Figure 5b) [29]. After that, the redundant Ka-band inter-satellite two-way clock can be used to estimate the clock offsets with higher accuracy (see Figure 5c) [19].

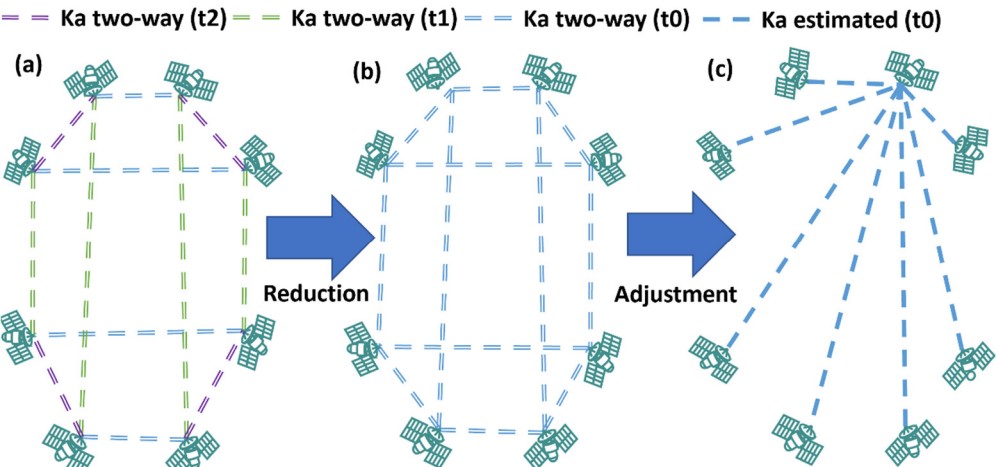

**Figure 5.** The procedure of the inter-satellite clock offset estimation. Wherein, t0, t1, t2 are different epoch within 60 s. Ka two-way is the Ka-band two-way clock offsets. Despite their epoch are different (as (**a**)), they can be reduced to the same epoch by prior message (as (**b**)) for further adjustment processing in the new strategy (as (**c**)).

After selecting a satellite as a reference satellite, "z." In the process of indirect adjustments, the observations are redundant Ka-band two-way inter-satellite clock offsets, and the parameters to be estimated are the Ka-band two-way inter-satellite clock offsets with respect to the reference satellite. The derivatives of observations of parameters to be estimated are given in Equation (2).

$$\begin{cases} \dfrac{\partial Clk^{Ka}{}_{AB}}{\partial \vec{x}} = \begin{bmatrix} 1, -1, 0...0 \end{bmatrix} \\ ... \\ \dfrac{\partial Clk^{Ka}{}_{Az}}{\partial \vec{x}} = \begin{bmatrix} 1, & 0,0...0 \end{bmatrix} \\ ... \\ \dfrac{\partial Clk^{Ka}{}_{zB}}{\partial \vec{x}} = \begin{bmatrix} 0, -1, 0...0 \end{bmatrix} \\ ... \end{cases} \qquad (2)$$

where, $\vec{x}$ are the clock offsets of all satellites with respect to a reference satellite, $Clk^{Ka}_{AB}$ are the Ka-band two-way inter-satellite clock offsets between the *A* and *B*. The results from the trial suggest that if the Ka-band two-way inter-satellite clock offsets within 1 min of the whole constellation are used to reduce, there will be more than 300 two-way links, and the reduced errors will be less than 7 mm. The comparison of the clock offsets before and after indirect adjustment is shown in Figure 6. The dark blue curves present the clock offsets after indirect adjustments, and the azury blue present the original Ka-band two-way inter-satellite clock offsets. Both of them deduct the same trends. From Figure 6. the advantages of the further indirect adjustments processing are that it not only can estimate inter-satellite clock offsets when they are not linked. Moreover, compared with Ka-band two-way inter-satellite clock offsets, the clock offsets, by further estimated, have higher accuracy and less noise.

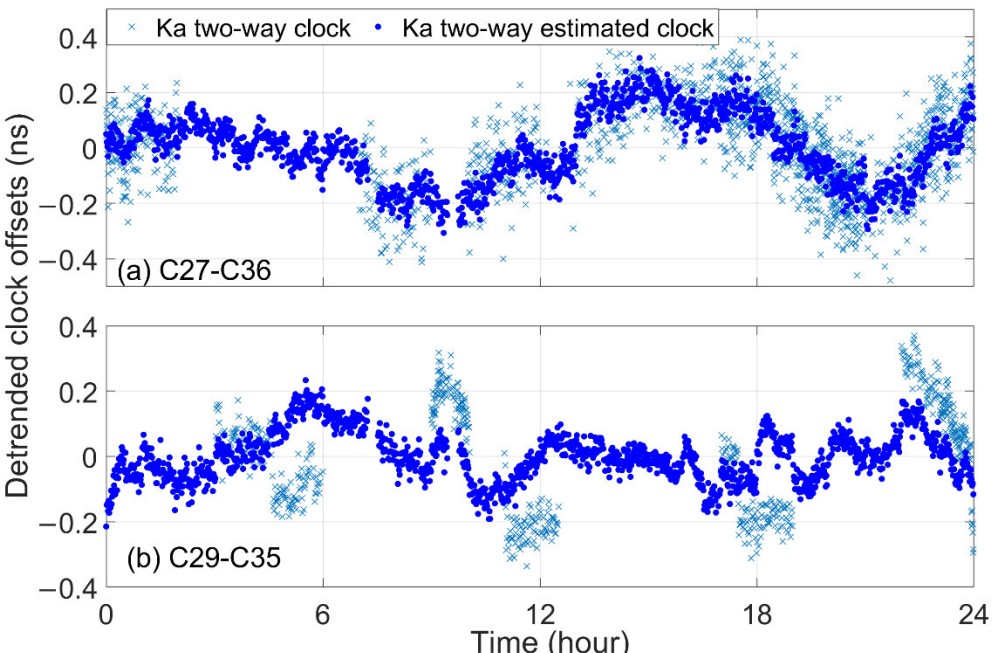

**Figure 6.** The improvements after using redundant Ka-band inter-satellite clock offsets to estimate Ka-band inter-satellite clock offsets (since 19 May 2022).

### 2.2.2. Estimation of Clock Offsets with Respect to BDT

The new strategy can estimate high-accuracy inter-satellite clock offsets by least square estimation processing for redundant Ka-band two-way inter-satellite clock offsets. However, the reference of satellite clock offsets parameter is BDT. The inter-satellite two-way estimated clock offsets need to be further traced to BDT. Despite no good ideas to directly solve the problems shown in Figure 1, we find that these problems can be effectively circumvented with some changes in the estimated clock offsets strategy. Firstly, the BDS-3 GEO is always over China, and MCS's antennas can track BDS-3 GEO anytime. BDS-3 can estimate GEOs clock offsets without passing through Ka-band two-way inter-satellite clock offsets and C-band two-way inter-station clock offsets. Then, the velocity of GEO related to antenna ground is slow, and the altitude angle and azimuth angle of tracked antenna nearly remain unchanged. Lastly, GEOs need to broadcast the messages of other services, including Satellite Based Augmentation System Service (SBAS) [30,31] and Precise Point Position Service in B2b signal (PPP-B2b) [32]. The above services require MCS to use antennas with a larger diameter and greater signal transmission power to track GEOs full time, which will help suppress the multipath effect in two-way observation. Therefore, in the satellite-ground clock offsets estimation, only the L-band two-way

clock offsets of the large-diameter antennas following GEOs are considered the satellite-ground clock offsets estimation.

Similar to the inter-satellite clock offsets estimation, in order to filter the noise and errors, the GEO's clock offsets estimation also adopts the least square estimation method. However, in the estimation, the zero value of different L-band antennas should be calibration and aligned. Set a GEO satellite as a reference "$Q$" the parameter to be estimated includes Q's clock offsets with respect to BDT, and others are the differences of zero value of L-band larger-diameter antennas. After that, the observation equations in the least square estimation are given as Equation (3).

$$\begin{cases} \rho_Q(t) = clk_Q^{L}(t) = clk_Q(t) \\ \rho_A(t) = clk_A^{L}(t) - clk_{AQ}^{Two-way\ estimated}(t) = clk_Q(t) + Bias_A^{L} \\ \rho_B(t) = clk_B^{L}(t) - clk_{BQ}^{Two-way\ estimated}(t) = clk_Q(t) + Bias_B^{L} \end{cases} \tag{3}$$

where the derivatives of observation of the parameters to be estimated are given as Equation (4).

$$\begin{cases} \dfrac{\partial \rho_Q}{\partial \vec{x}} = [(0,0)_{\vec{x}_0}, (1)_{\vec{x}_1}] \\ \dfrac{\partial \rho_A}{\partial \vec{x}} = [(1,0)_{\vec{x}_0}, (1)_{\vec{x}_1}] \\ \dfrac{\partial \rho_B}{\partial \vec{x}} = [(0,1)_{\vec{x}_0}, (1)_{\vec{x}_1}] \end{cases} \tag{4}$$

wherein, $clk_A^{L}(t)$, $clk_B^{L}(t)$, $clk_Q^{L}(t)$ are the L-band two-way clock offsets of three GEO satellites. $clk^{Two-way\ estimated}$ are the Ka-band two-way inter-satellite estimated clock offsets. are the parameters to be estimated, which include the global parameter $\vec{x}_0$ and local parameter $\vec{x}_1$. Global parameters are the zero value differences, including the $Bias_A^{L}$ and $Bias_B^{L}$. Local parameters include the clock offsets of the reference satellite at different epochs. In the least square estimation processing, the global and local parameters can be effectively separated by operating the normal equation to eliminate parameters. Thus, the update of the antenna's bias and the estimation of higher-accuracy clock offsets of BDS-3 GEOs can be both real-time realized. The RMS of the 9-day residual for the least square estimation is 0.1 ns, which are given in Figure 7. Figure 7 preliminary suggests that the observations of different large-diameter antennas present a good consistency in short timescale. Their long-time consistency needs to be further verification in the future.

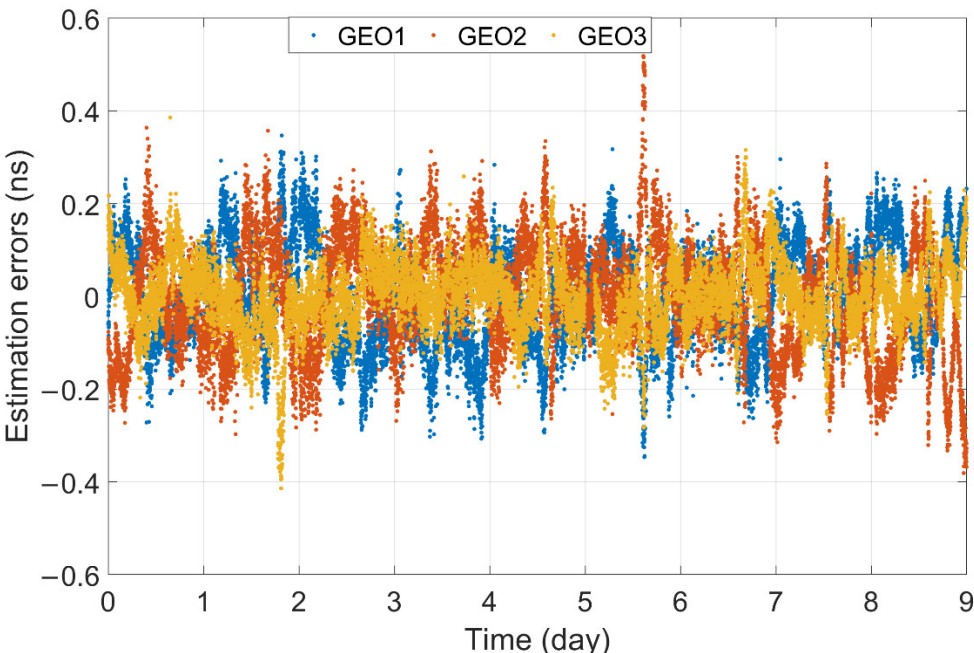

**Figure 7.** The estimation errors of the second least square estimation (since 19 May 2022), where the reference satellite is GEO-3.

Considering the high-accuracy inter-satellite clock offsets already obtained by 3.1. the clock offsets of all other satellites can be further traced to BDT by Equation (5).

$$clk_A(t_0) = clk_Q(t_0) + clk^{Two-way\ estimated}_{AQ}(t_0) \tag{5}$$

### 2.2.3. Estimation and Elimination of the Bias of Two-Way Clock Offsets

By the method in Sections 2.2.1 and 2.2.2, the new strategy can estimate the two-way clock offsets with higher accuracy in real-time. However, bias still exists between the two-way estimated clock offsets and ODTS clocks offsets. This bias includes the zero value of signal transmitting and receiving of Ka-band antennas and L-band antennas. However, BDS-3 users use one-way downlinks observation for positioning and timing. Therefore, to align the two-way estimated clock offsets with the one-way downlinks signal, the new strategy should finally correct the bias in the two-way estimated clock offsets.

The effective way to estimate the bias is using ODTS clock offsets. The duration of BDS-3 orbit determination is three days, and the specific strategy of BDS-3 orbit determination can refer to Tang et al. In order to suppress the influence of ODTS clocks offset errors on the bias estimation. The new strategy uses multiple-day ODTS clock offsets to estimate the bias. As shown in Figure 8, the estimation and elimination of the bias of the two-way estimated clock offsets include three steps. Firstly, as shown in Figure 8, splice the original ODTS clock offsets of multiple days to obtain the ODTS clock offsets (at present, the number of days is 3, which are consistent with the two-way superimposition strategy). Secondly, make the difference between the ODTS clock offsets and two-way estimated clock offsets, and average their differences to estimate the bias. Thirdly, deduct the bias in two-way estimated clock offsets so as to realize the alignment of the two-way clock offsets with the one-way downlink signal.

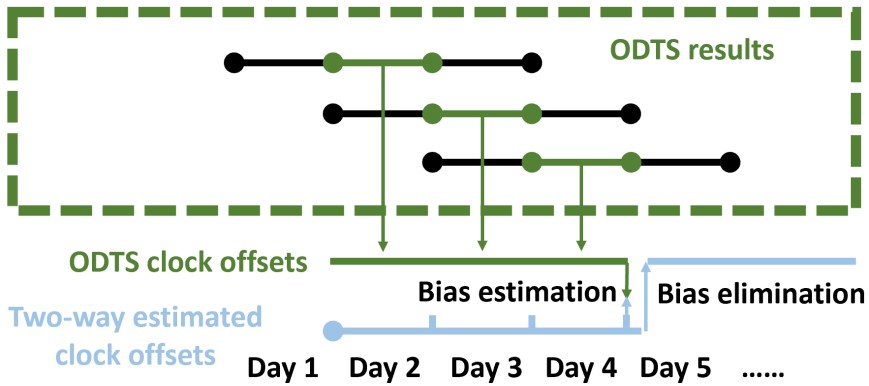

**Figure 8.** The procedure of the estimation and elimination of the bias of the two-way estimated clock offsets.

## 3. Results

In the new strategy, only using redundant ISL two-way observations and first least square estimation processing, the variations of inter-satellite clock offsets between other satellites can be real-time estimated. The two-way inter-satellite clock offsets are independent of L-band two-way measurements. Taking the clock offsets between the MEO-1 and MEO-2 as the example, the differences in the 1-day quadratic fitting residual between the two-way inter-satellite estimated clock offsets and the ODTS inter-satellite clock offsets from the German research center for geosciences Beidou multi-GNSS (GBM) [33] are plotted in Figure 9. Wherein, the blue curves of Figure 9a are the two-way estimated clock offsets between the MEO-1 and MEO-2. The green curves of Figure 9a are their ODTS clock offsets. Both of them are deducted with the same trends, and their differences are plotted in Figure 9b. Although the STD of their differences is less than 0.1 ns in the time-scale of 1 day, due to the day-boundary discontinuities in the ODTS clock offsets [34], their differences exist in the obvious "hop" at the boundary between two days, which are given in Figure 10.

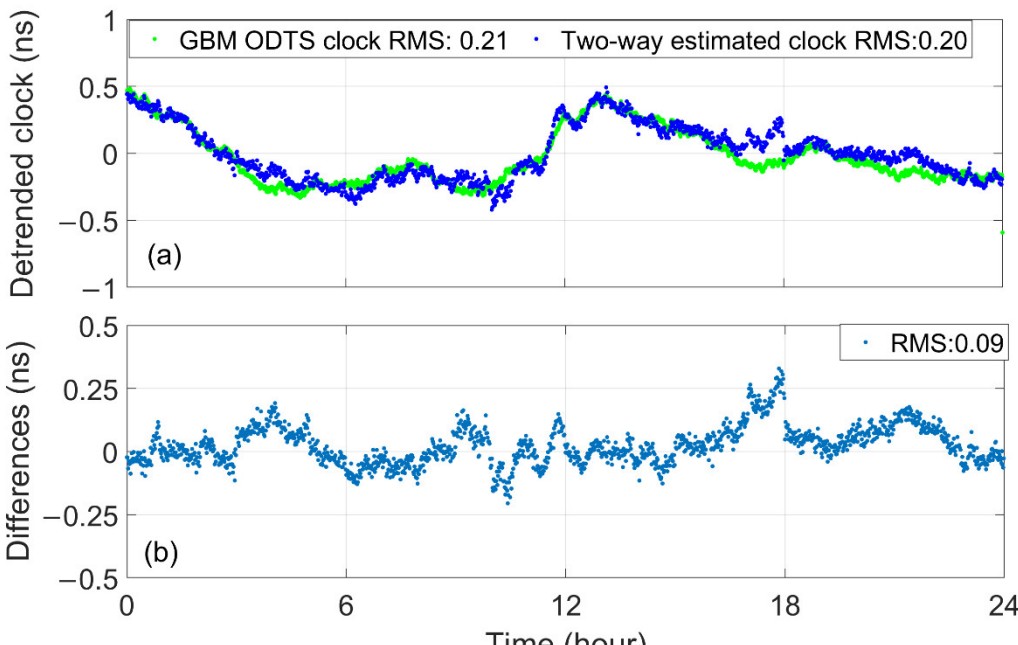

**Figure 9.** 1-day Satellite clock offset differences between the Ka-band inter-satellite estimated clock offsets and GBM products (since 19 May 2022). Wherein, (**a**) describes the two kinds of clock offsets after deducting the same trend. (**b**) describes their differences.

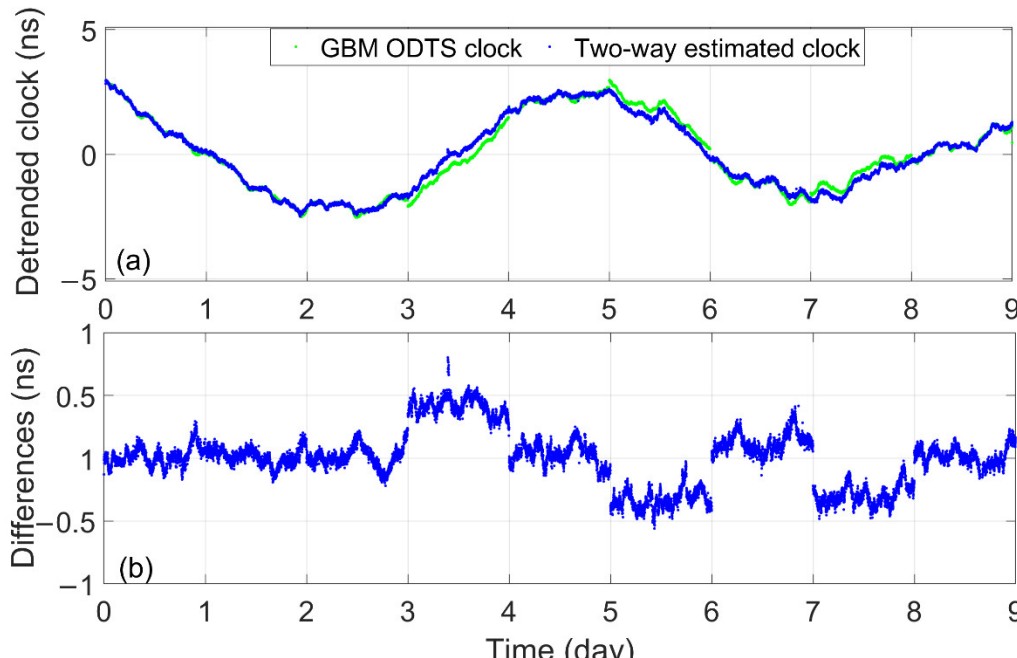

**Figure 10.** 9-day Satellite clock offset differences between the Ka-band inter-satellite estimated clock offsets and GBM products. Wherein, (**a**) describes the two kinds of clock offsets after deducting the same trend. (**b**) describes their differences. The reason why "hop" exist in the bottom figure is that the day-boundary discontinuities exist in the ODTS clock offsets (since 19 May 2022).

In order to suppress the influence of day-boundary discontinuities in the ODTS clock offsets, we deducted the same trends per day in the following comparisons between the ODTS clock offsets and the two-way clock offsets. Referring to BDS-3 M1, Tab1 lists 9-day averaging statistical results of their 1-day quadratic fitting residual and the STD of their differences for all BDS-3 satellites. Except for BDS-3 C61, the statistical results show that the averaged STD of the differences between the two-way estimated clock offsets and GBM clock offsets is 0.19 ns. Among them, the averaged STD for BDS-3 MEOs is 0.14 ns and 0.40 ns, 0.36 ns for IGSOs and GEOs, respectively.

As shown in Table 1, In most cases, the large differences between the two kinds of clock offset with the larger fitting residuals in the GBM ODTS clock offsets. Epically for the GEOs and IGSOs. Taking the two kinds of clock offset between the IGSO-1 and MEO-1 as an example, their comparisons are given in Figure 11. Both of them are deducted the same trends per day. From Figure 10. It can be clearly found that their larger differences are caused by the large fluctuation in GBM ODTS clock offsets. We believe that the IGSOs and GEOs clock offsets with large fluctuations are that there exist larger orbit errors in GEOs and IGSOs. Some results suggest it is probably due to the not accurate enough solar radiation pressure model adopted in ODTS processing [33,35].

**Table 1.** Averaging statistical results of the two kinds of 1-day clock offsets comparison.

| Satellite PRN | The STD of the Differences between Two Kinds of Clock Offsets (ns) | 24 h Fitting Residual (ns) | |
| --- | --- | --- | --- |
| | | ODTS Clock (GBM) | Two-Way Estimated Clock |
| C38 | 0.25 | 0.26 | 0.12 |
| C39 | 0.36 | 0.37 | 0.09 |
| C40 | 0.39 | 0.39 | 0.08 |
| C25 | 0.16 | 0.18 | 0.11 |
| C26 | 0.16 | 0.14 | 0.09 |
| C27 | 0.11 | 0.09 | 0.11 |
| C28 | 0.13 | 0.13 | 0.12 |
| C29 | 0.19 | 0.13 | 0.22 |

| | | | |
|---|---|---|---|
| C30 | 0.15 | 0.17 | 0.10 |
| C20 | 0.07 | 0.18 | 0.16 |
| C21 | 0.09 | 0.21 | 0.21 |
| C22 | 0.07 | 0.18 | 0.17 |
| C23 | 0.14 | 0.18 | 0.14 |
| C24 | 0.12 | 0.16 | 0.17 |
| C32 | 0.11 | 0.12 | 0.12 |
| C33 | 0.12 | 0.18 | 0.14 |
| C34 | 0.15 | 0.13 | 0.08 |
| C35 | 0.19 | 0.19 | 0.10 |
| C36 | 0.14 | 0.13 | 0.07 |
| C37 | 0.11 | 0.15 | 0.10 |
| C41 | 0.08 | 0.10 | 0.10 |
| C42 | 0.11 | 0.12 | 0.11 |
| C43 | 0.15 | 0.24 | 0.15 |
| C44 | 0.12 | 0.20 | 0.16 |
| C45 | 0.23 | 0.26 | 0.08 |
| C46 | 0.23 | 0.25 | 0.11 |
| C59 | 0.50 | 0.65 | 0.32 |
| C60 | 0.61 | 1.00 | 0.17 |
| MEO | 0.14 | 0.19 | 0.13 |
| IGSO | 0.40 | 0.71 | 0.11 |
| GEO | 0.36 | 1.10 | 0.36 |
| Average of all satellites | 0.19 | 0.23 | 0.13 |

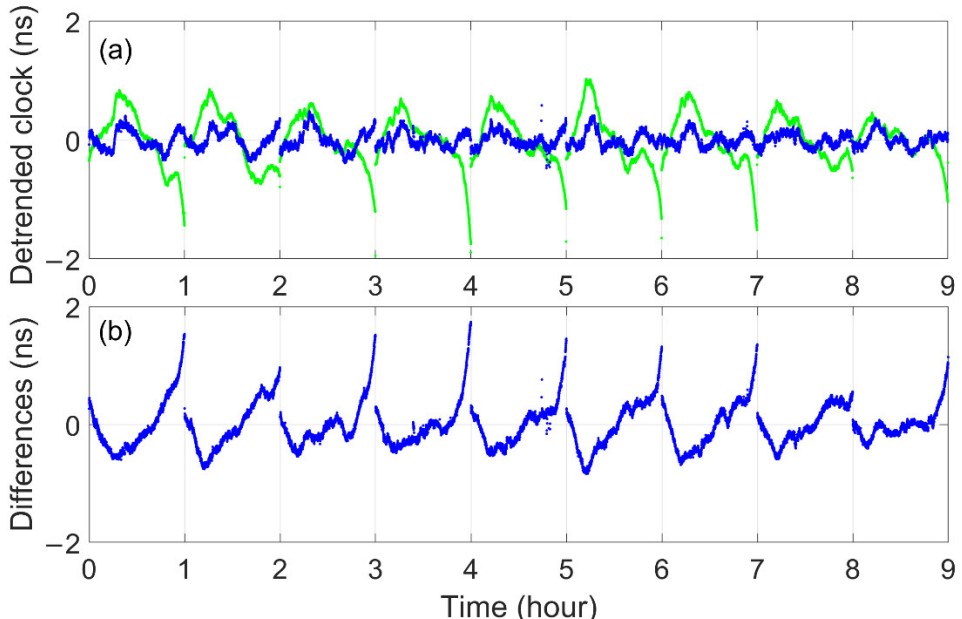

**Figure 11.** Satellite clock offset differences between the Ka-band inter-satellite estimated clock offsets and GBM products for 9 days. All of them are deducted the same trends per day (since 19 May 2022). Wherein, (**a**) are their detrended clock offsets, (**b**) are their differences.

Taking the high-accuracy inter-satellite clock offsets and L-band two-way clock offsets of GEOs as inputs, the clock offsets of all satellites can be further traced to the BDT so as to estimate the final clock offsets, whose accuracy should be better than the clock offsets calculated by two-way superimposition strategy, which can be shown in Figure 9. The top panel of Figure 9 describes the GEO clock offsets. The middle panel of Figure 9 describes the typical IGSO clock offsets, and the bottom panel of Figure 12 describes the typical MEO clock offsets curves are the two-way estimated clock offsets. Wherein the red curves are the clock offsets directly calculated by L-band two-way clock offsets or superimposed by L-band two-way clock offsets and C-band inter-satellite clock offsets.

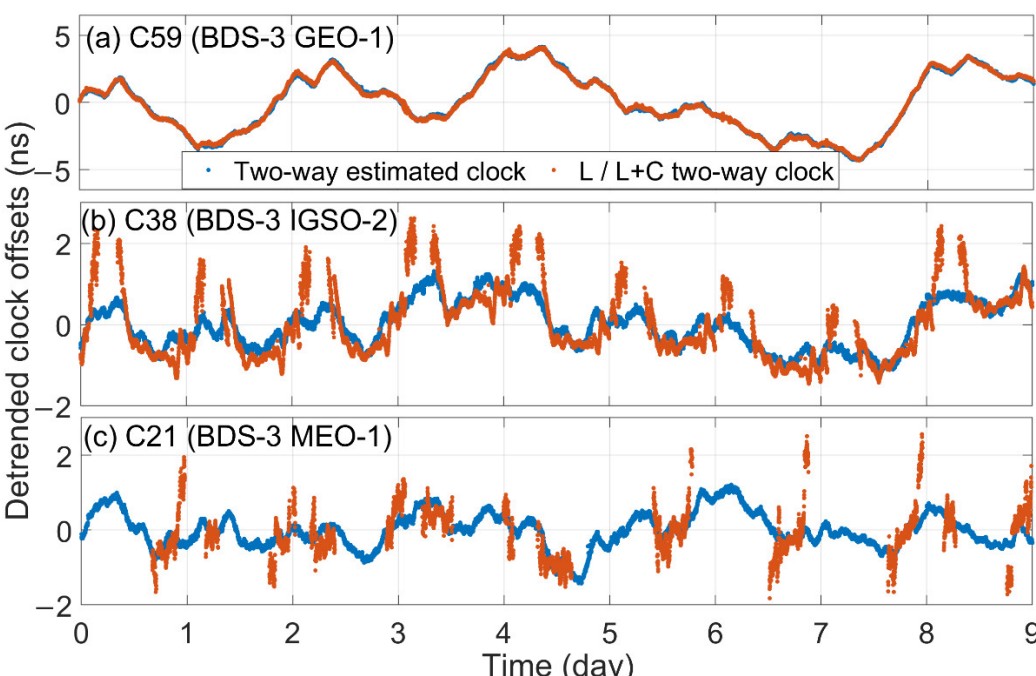

**Figure 12.** Satellite clock offset differences between the two-way estimated clock offsets and the two-way intraregional satellite clock offsets measured by superimposition strategy (since 19 May 2022).

By comparing the two curves, it can be found that although the red curves are in good agreement with the blue curves in most cases, the red curves become more divergent than the blue curves at the beginning and end of each segment, which indicate the two-way estimated clock offsets have more advantages than two-way clock offsets. The above phenomenon is because the new strategy effectively solves problems in the two-way superimposition strategy:

- Because the new strategy uses the least square estimation method to estimate clock offsets and will not consider C-band two-way inter-station clock offsets with large noise, the clock offsets can always keep higher precision.
- The final two-way clock offsets between the satellite atomic clock onboard and BDT are only realized through the two-way observation between the MCS larger diameter antennas and GEOs, and the elevation of antennas tracking to GEOs is unchanged all the time. The problem of antennas with low elevation no longer exists.
- The new strategy only introduces three larger-diameter L-band antennas, each of which tracks to the specific GEO anytime. On this basis, the new strategy introduces the bias parameters to absorb the inconsistencies of observations of larger-diameter L-band antennas. All of the above considerations are able to suppress the "step errors."
- In the new strategy, the concept of intraregional satellite and overseas satellite no longer exist. All satellites play the same role in the inter-satellite clock offsets estimation. Furthermore, their clock offsets with respect to BDT are not dependent on ISL plans and antenna status but can be calculated through the same clock offsets (reference GEO). Therefore, even if IGSOs and MEOs are not able to be observed by ground antenna, the accuracy of their clock offsets will not be lost.

After obtaining the satellite clock offsets with higher accuracy, the BDS-3 is able to generate the satellite clock offsets parameters with higher accuracy. In most cases, BDS-3 satellite clock offsets parameters are generated by the satellite clock offsets of 2 hours in history. Moreover, their prediction duration is less than 2 hours. The Table 2. shows the statistical results of 2-hour prediction errors of two-way estimated clock offsets for 9-day. From Table 2, the averaged RMS of all satellites is 0.29 ns, which is 29% of those calculated

by the BDS-3 two-way superimposition strategy. Considering the clock offsets parameters errors of Galileo is 0.14 m (0.42 ns), the above results indicate that after using the new clock offsets estimation strategy, the accuracy of BDS-3 clock offsets parameters could reach or exceed those of Galileo.

**Table 2.** Statistical results of 2-hour prediction errors of two-way estimated clock offsets for 10 days.

| PRN | RMS (ns) | 95% Error (ns) |
|---|---|---|
| C19 | 0.27 | 0.55 |
| C20 | 0.27 | 0.58 |
| C21 | 0.31 | 0.66 |
| C22 | 0.31 | 0.67 |
| C23 | 0.30 | 0.61 |
| C24 | 0.29 | 0.62 |
| C25 | 0.28 | 0.59 |
| C26 | 0.29 | 0.65 |
| C27 | 0.26 | 0.56 |
| C28 | 0.28 | 0.62 |
| C29 | 0.30 | 0.64 |
| C30 | 0.27 | 0.59 |
| C32 | 0.27 | 0.57 |
| C33 | 0.28 | 0.59 |
| C34 | 0.28 | 0.59 |
| C35 | 0.28 | 0.59 |
| C36 | 0.27 | 0.56 |
| C37 | 0.25 | 0.53 |
| C38 | 0.28 | 0.62 |
| C39 | 0.26 | 0.53 |
| C40 | 0.26 | 0.56 |
| C41 | 0.29 | 0.65 |
| C42 | 0.26 | 0.54 |
| C43 | 0.28 | 0.57 |
| C44 | 0.37 | 0.77 |
| C45 | 0.27 | 0.58 |
| C46 | 0.3 | 0.62 |
| C59 | 0.33 | 0.71 |
| C60 | 0.36 | 0.86 |
| C61 | 0.38 | 0.66 |
| Average of all satellites | 0.29 | 0.61 |

## 4. Conclusions

In order to improve the signal-in-space accuracy of the BDS-3 satellite, the paper introduces the two-way superimposition strategy and analyzes its problems in satellite clock offsets measured. The analysis shows that BDS-3 uses the two-way superimposition strategy that can effectively calculate the clock offsets, both including intraregional satellite and overseas, with respect to BDT in real-time. Nevertheless, suffering from antenna zero value error, the low elevation angle of antennas, and the large noise of C-band interstation clock offsets, the clock offsets calculated by BDS-3 are not accurate enough. This problem is the main factor limiting the improvement of space-in-signal accuracy for BDS-3.

Based on the above analysis, the paper optimizes and updates the BDS-3 strategy in BDS-3 satellite clock offsets estimation. Unlike the simple superposition of two-way observations, the new strategy from the perspective of least square estimation takes full advantage of the BDS-3 multi-source observation and the BDS-3 constellation characteristics to circumvent problems in the two-way superimposition strategy so as to improve the accuracy of the BDS-3 satellite clock offsets. The satellite clock offsets estimated by the new strategy have the following characteristic:

- The two-way estimated clock offsets retain the advantages of two-way clock offsets, which are less affected by propagation and position errors than one-way downlink clock offsets.
- All ISLs maintain the inter-satellite relative variation of two-way estimated clock offsets. The inter-satellite relative variation of two-way estimated clock offsets is independent of satellite-ground links.
- All satellite clock offsets are traced to BDT by the same clock offsets, which are only affected by the observations of three large-diameter L-band antennas.

By optimizing and updating, the BDS-3 are able to improve the accuracy of not only BDS-3 inter-satellite clock offsets but also the final BDS-3 satellite clock offsets with respect to BDT. Making the differences between the clock offsets estimated by the new strategy and the GBM products, the average STD of their clock offsets is 0.19 ns, wherein MEO is 0.14 ns. Furthermore, compared with the inter-satellite clock offsets of GBM products, the 24-h fitting residual of two-way estimated clock offsets is more stable. The above results indicate that: Only relying on ISLs and a few L-band large-diameter antennas, the BDS-3 can realize the global satellite clock offsets real-time estimation with the same accuracy level as the clock offsets estimated by more than one hundred global-distributed GNSS stations [33,35].

After adopting the new strategy, the accuracy of the satellite clock offsets parameter can be improved from 1.05 ns (0.35 m) [8] to 0.29 ns (0.09 m), about 71%. This result indicates the ISL can not only solve the problems of insufficient observation in overseas satellite clock offsets estimation but also can enhance the accuracy of satellite clock offsets parameters of intraregional satellites. The signal-in-space errors of BDS-3 broadcast ephemeris are 0.09 m. Considering that the SISRE are less than the sum of the broadcast ephemeris errors and clock offsets parameter errors [8], after adopting the proposed strategy in the BDS-3 time synchronization system, the signal-in-space accuracy of the BDS-3 navigation message can be less than 0.2 m at least. This result is of great significance to the performance improvement of BDS.

As shown in Table 2, after BDS-3 used new strategy to estimate the clock offsets, a puzzling discover is that despite the performance of all BDS-3 clock offsets are different, there is little different between their prediction ability. The reason is plotted in Figure 13. Multi-day fitting residuals of all satellite with the highest-level-performance atomic clock are present the almost common fluctuation, and the same fluctuation is found in the output signals of BDT. Although the common fluctuation will not affect the final position accuracy, it will affect the timing accuracy. It indicates that the common fluctuation in Figure 13 is caused by the fluctuation of BDT itself [2]! Therefore, the stability of the BDT's output signal limits the ability for BDS-3 to improve further the accuracy of BDS-3 satellite's clock offset parameters.

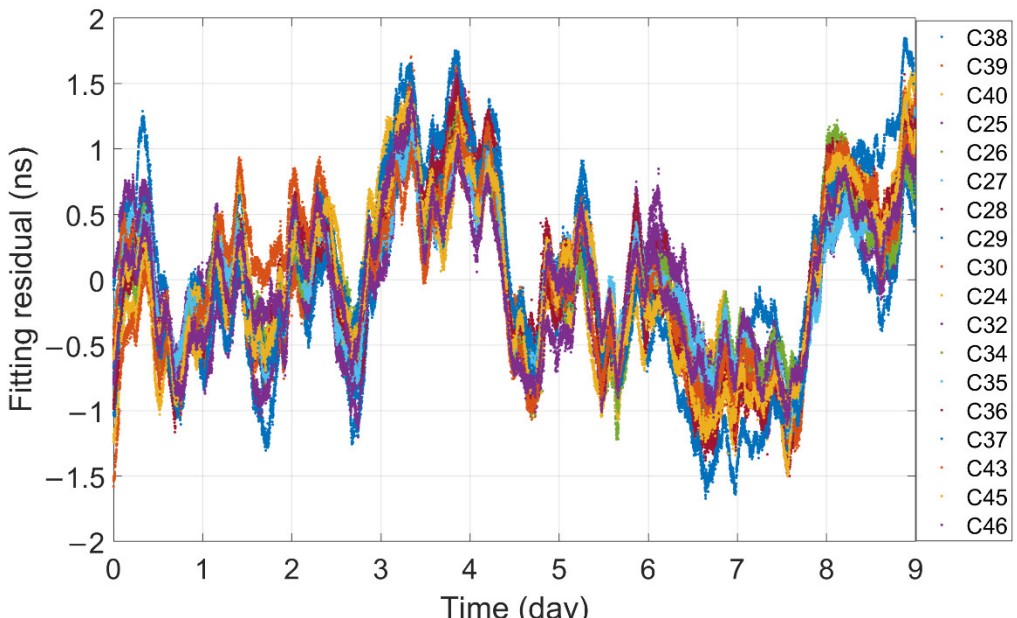

**Figure 13.** Multi-day fitting residuals of BDS-3 satellite with highest-level-performance atomic clock onboard with respect to BDT (since 19 May 2022). The common fluctuation is caused by the fluctuation in BDT output signals.

It should be noticed that, despite the above results is inspiring for BDS-3, from the perspective of service stability, BDS-3 still need longer time to check this strategy. What is more, in the bias estimation of the two-way estimation clock offsets, the inputs ODTS clock offsets are dual-frequency ionospheric free combination clock offsets (usually B1I/B3I). Therefore, the final two-way clock offsets are aligned with the one-way downlinks clock offsets with the specific combination frequency. However, for the users of other BDS-3 frequencies. They need to use the broadcast DCB and TGD, and ionospheric parameters to further correct their observation. However, because of their strong correlation and poor distribution of BDS-3 GNSS stations, it is also tricky for BDS-3 to realize their high-precision calculation under the condition that GNSS stations of BDS-3 are only distributed within China. Therefore, after optimizing and updating the clock offsets estimation strategy, the accuracy of the parameters of ionospheric and DCB/TGD will become the main point of growth of BDS-3 accuracy. Improving their accuracy is another key technology that BDS-3 needs to further breakthrough.

Finally, we can see a good agreement between the one-way downlink clock offsets and the two-way estimated clock offsets, especially for the MEO satellites with higher ODTS clock offsets accuracy. It probably means that the unknown system errors between the ISL and L-band downlinks are not large. Considering that the two-way estimated clock offsets with less fluctuation than the current ODTS clock offsets in the timescale from the few hours to 1 day. We believe that if BDS-3 takes the two-way estimated clock offsets as the new observation in the GNSS orbit determination, BDS-3 will be able to enhance the accuracy of orbit determination further and accelerate the decouple and convergence time of GNSS parameters, which will be conducive to the study in other GNSS fields [20–22] and the improvement of the ability on BDS-3 PPP-B2b [32]. All of them deserve further study.

**Author Contributions:** J.Y., C.T., Y.Y., and X.H.; Data curation, Y.X. and Z.Y.; Formal analysis, J.L.; Funding acquisition, S.Z.; Investigation, Y.S., Y.L., and Z.Y.; Methodology, J.Y., S.Z., and Y.S.; Project administration, X.H.; Resources, X.H.; Software, J.Y. and Y.S.; Supervision, Y.X., Y.Y., and X.H.; Validation, J.L., Y.X., Y.L., and Z.Y.; Visualization, J.Y.; Writing—original draft, J.Y.; Writing—

review & editing, J.Y., Q.T., and X.H. All authors have read and agreed to the published version of the manuscript.

**Funding:** This research was supported by Shanghai Observatory's key cultivation project (N20210601003); National Natural Science Foundation of China (No. 12173072); Civil Aerospace "14th Five-Year" Technology Pre-research Project (KJSP2020020203).

**Data Availability Statement:** The data used in this contribution, including the ISL measurements, L-band two-way measurements, C-band two-way measurements, are provided by the Beijing Satellite Navigation Canter. The above data are available from the corresponding author upon request. The other data used in this contribution include the BDS-3 broadcast message provided by the Test and Assessment Research Center of the China Satellite Navigation Office and the precise clock data provided by the GBM, which can be downloaded from the internet.

**Conflicts of Interest:** The authors declare no conflict of interest.

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
