# Peer review of "High-Accuracy Clock Offsets Estimation Strategy of BDS-3 Using Multi-Source Observations"

_remotesensing, doi:10.3390/rs14184674_

Round 1
Reviewer 1 Report
The article entitled: High-accuracy clock offsets estimation strategy of BDS-3 using 2 multi-source observations presents a new clock offsets estimation for BDS satellites. This is an interesting and important research result that can directly improve the time synchronization accuracy of BDS-3 satellites, and thus improve the PVT performance for civil BDS users. Although there are still some unknown systematic biases (e.g., antenna phase center, Ka pseudorange bias) in the use of Ka intersatellite links themselves, the impact of such systematic biases is not significant anymore as can be seen from the comparison with GBM's products. The results of the study are very meaningful for improving the service performance. Therefore, the topic is of considerable interest and the paper is analyzed in original submission. In my opinion, this paper can be published after the minor modifications below:
1. All chapters are very well described and they do not raise any doubts. In terms of the literature review is not complex, most of which are papers from recognized scientific journals, such as: GPS Solutions, Journal of Geodesy, and others. However, I would like to point out that in the introduction section, the authors are not sufficiently aware of the latest research results about BDS-3 signal in space performance evaluation, and suggest adding the corresponding references appropriately.
2.P1, L43-44, the clock offsets parameters errors of BDS-3 are 0.35m (about 1.05ns)
You need to point out what the unit is? RMS or STD?
3.P2, L59, Fig1 should be Fig.1.
4.P2, L77, Please indicate which of the different colors in the diagram indicates the C-band, Ka-band, and L-band two-way measurements
5.P15, L402, on the few L-bond or on a few L-bond?
6.P15, L410-412,In this part, the authors do not discuss and calculate, and I did not find relevant supporting materials
Author Response
Thanks for your recognition of our works, and thanks for your constructive suggestions and helpful comments. And we are so grateful to you for pointing out many weaknesses of the manuscript. We attach great importance to each of your suggestions, and each of them are replied in the following responses. Each relevant changes in the revision are marked in red in the subsequent replies. If there exist any question in the reply or new manuscript, we would appreciate it if you could tell us.

Reviewer 2 Report
Manuscript Number: remotesensing-1896136
Full Title: High-accuracy clock offsets estimation strategy of BDS-3 using multi-source observations
General comment
The article submitted to Remote Sensing MDPI by Xiaogong Hu et al. studied accuracy of clock offsets estimation strategy of BDS-3, using multi-source observations. The paper is analyzed in original submission and is structured in: 1. Introduction, 2. Two-way superimposition strategy, 3. The new strategy, 4. Accuracy and improvements, 5. Conclusions and discussion, References.
From a first reading-review, it appears to this reviewer which has not been edited according to the RS MDPI instructions for authors; with this structure, it is difficult to read and understand the methods, analyse the results, evaluate the discussion of them.
For this reviewer is unambiguous, however, that the study shows a considerable effort on the part of authors to implement the project and its potential applications. The content of the paper is potentially interesting, and with improvement, it might be published.
However, in its current form it is not acceptable, in my opinion (especially for a high quality journal like RS MDPI) for several reasons, that follow in the specific comments.
Specific comment
With regret, the paper shows some limitations that need to be deeply resolved by the authors with a new and completely revised version of the manuscript.
In particular, the authors must follow these recommendations:
1. read the authors' instructions and set up the paper according to the following scheme: Introduction, Materials and Methods, Results, Discussion, Conclusions. There is no firm standard way of writing an article, but the article should be readable by people in this field;
2. the section Introduction must be revised, discussing the other articles in the topic, with particular reference also to the general ones concerning BDS3 and Multi GNSS;
3. the section 2 and 3 should be merged in “Materials and Methods”;
4. the section 4 should be renamed in “Results”;
5. the section 5 must be inserted with “Discussion” (is mandatory in RS…);
6. the authors must demonstrate to this reviewer and the reader that using data for only nine days produces results that are scientifically sound. In my opinion this dataset is too short, it needs to be extended to at least one month;
7. last but not least, the references are too limited and need to be extended, for a work to be reported in one of the most prestigious journals like RS MDPI.
In conclusion the work deserves publication after major revisions, in my opinion. I will be available to authors for the assessment of the manuscript at the subsequent submission, I summarize in technical comment below.
Best regards
Technical comment
1. line 31, the section Introduction must be revised, discussing the other articles in the topic, with particular reference also to the general ones concerning BDS3 and Multi GNSS, I suggest (but it is not mandatory): DOI: 10.1007/s00190-008-0300-3, DOI: 10.1016/j.asr.2017.01.011, DOI: 10.1007/s00190-015-0802-8, https://doi.org/10.3390/rs14163930; 2. line 58, please move figure 1 to Materials and Method; 3. line 77, please see previous comment 2; 4. line 85, please renamed Two-way superimposition strategy with Materials and Method, for the reader; 5. line 110. “According to the antenna track messages and ISL plans”, please insert a reference for this; 6. line 121, after Figure 3, please modify text from bold to standard; 7. line 151, The new strategy is still “Materials and method”, please modify; 8. line 187, please insert “.” After Figure 5; 9. line 206, figure 6, please modify the figure, by inserting the x-axis (I assume hours); 10. line 222, Please chech “it”; 11. line 247-249, The authors said: “The RMS of the 9-day residual for the least square estimation is 0.1ns,which are given in Fig.7. This result suggests that the observations of different large-diameter antennas present a good consistency”. The authors must demonstrate to this reviewer and the reader that using data for only nine days produces results that are scientifically sound. In my opinion this dataset is too short, it needs to be extended to at least one month; 12. line 264, now are three day, why? Please specify in the text; 13. line 278, please insert “Results”; 14. line 297, please see previous comment 8; 15. line 308, only “Average” is showed in the table, please justify this for the reader; 16. line 319, please see previous comment 8; 17. line 331, please see previous comment 8; 18. line 367, please see previous comment 15; 19. line 371, insert here “Discussion” (is mandatory in RS…); 20. line 425, please see previous comment 8; 21. line 472, the references are too limited and need to be extended, for a work to be reported in one of the most prestigious journals like RS MDPI. For example see: DOI: 10.1186/s43020-019-0006-0, DOI: 10.1002/navi.295, DOI: 10.1007/s10291-019-0928-x, DOI: 10.3390/rs12132081;22. references 17 and 23 are not published, please delete.
Author Response

(The authors gave the same response as above.)

Reviewer 3 Report
The work “High-accuracy clock offsets estimation strategy of BDS-3 using multi-source observations” seems to be interesting but, to me, it is partly unclear with regard to a new methodology and its comparison with the older one. The authors provide diagrams (Figures 2 and 4) but without mathematical models, it is difficult to understand the novelty. Please note that looking at the mentioned figures you just reduced the number of input data removing C-band two-way observations. The entire should be as clear as possible. The other issues that need improvement are:
1. Some sentences like “Therefore, the main reason why BDS-3 signal-in-space accuracy is inferior to Galileo is that BDS-3 shows terrible accuracy of clock offsets parameters” are not understandable. Please read the entire work carefully
2. Lines 64-65 “According to the tracking status of the ground antenna to satellite and inter-satellite link (ISL) plans” are not clear.
3. Line 70. What do you mean by “addition”?
4. Figure 1. If you highlight links with different colours, please explain them in the caption.
5. Point 2.1 describe the entire strategy two-way superimposition strategy with one basic equation. There is no information on how different data are combined in one mathematical adjustment model. To this point, Figure 3 clarifies the differences between the three different solutions but my question is related to the availability of the dataset. Is it (for example L+C+Ka) continuously available, i.e. for entire days?
6. Figure 5 and its description are not clear. What are t1,t2…?
7. Lines 196-200. It is not clear to me how any “indirect” adjustment of KA data provides more reliable results. Is the number of KA data the same in the old and new strategies?
8. Figure 6 and further. What days are used? Please specify dates.
9. The results given in Figures 8 and 9 suggest that both clocks (GBM and two-way estimated) are quite consistent, but table 1 reveals significant differences. Please discuss this effect more deeply.
10. What is the reason for the strong daily pattern for two-way estimated clocks (Figure 10 top panel)
11. Line 324, It is probably Figure 11 not 9
Author Response

(The authors gave the same response as above.)

Round 2
Reviewer 2 Report
Dear Authors,
the new version is improved considerably, so I accept in this form.
Best regards
Reviewer 3 Report
I am satisfied with the responses to my review. In my opinon the work can be accepted.